# Parallel Interconnected Kinetic Asymmetric Transformation (PIKAT) with an Immobilized ω-Transaminase in Neat Organic Solvent

**DOI:** 10.3390/molecules25092140

**Published:** 2020-05-03

**Authors:** Wesley Böhmer, Lucien Koenekoop, Timothée Simon, Francesco G. Mutti

**Affiliations:** Van ‘t Hoff Institute for Molecular Sciences, HIMS Biocat, University of Amsterdam, 1098 XH Amsterdam, The Netherlands; wbohmer91@gmail.com (W.B.); lucien.koenekoop@gmail.com (L.K.); timothee.simon@outlook.com (T.S.)

**Keywords:** biocatalysis, asymmetric catalysis, kinetic resolution, α-chiral amines, enzyme immobilization, EziG, cascades, PIKAT, transaminases, aminotransferases

## Abstract

Comprising approximately 40% of the commercially available optically active drugs, α-chiral amines are pivotal for pharmaceutical manufacture. In this context, the enzymatic asymmetric amination of ketones represents a more sustainable alternative than traditional chemical procedures for chiral amine synthesis. Notable advantages are higher atom-economy and selectivity, shorter synthesis routes, milder reaction conditions and the elimination of toxic catalysts. A parallel interconnected kinetic asymmetric transformation (PIKAT) is a cascade in which one or two enzymes use the same cofactor to convert two reagents into more useful products. Herein, we describe a PIKAT catalyzed by an immobilized ω-transaminase (ωTA) in neat toluene, which concurrently combines an asymmetric transamination of a ketone with an anti-parallel kinetic resolution of an amine racemate. The applicability of the PIKAT was tested on a set of prochiral ketones and racemic α-chiral amines in a 1:2 molar ratio, which yielded elevated conversions (up to >99%) and enantiomeric excess (*ee*, up to >99%) for the desired products. The progress of the conversion and *ee* was also monitored in a selected case. This is the first report of a PIKAT using an immobilized ωTA in a non-aqueous environment.

## 1. Introduction

ω-Transaminase enzymes (ω-TAs) display high reactivity and exquisite chemo-, regio- and stereo-selectivity under mild reaction conditions and are frequently applied as catalysts for the synthesis of α-chiral amines [1,2,3,4,5,6,7,8]. Among the many enzymes that have been applied for the synthesis of optically active compounds, ω-TAs can enable both asymmetric synthesis from prochiral ketones and kinetic resolution of racemic amines [1,2,3,4,5,6,7,8]. The enzymatic transamination reaction proceeds via the pyridoxal 5′-phosphate cofactor (PLP), which performs the reversible transfer of an amine group from an amine donor to a carbonyl compound acceptor. Whereas α- and β-transaminases are used for the synthesis of α- and β-amino acids, respectively, ω-TAs are distinguished from other members of the transaminase sub-class by their ability to convert substrates that are devoid of a carboxylate group, thereby making them attractive catalysts in chemical synthesis [7].

During the past decade, the applicability of ω-TAs in organic synthesis has been tremendously enhanced in terms of substrate acceptance and optimization of the reaction conditions to solve (or at least alleviate) challenging issues such as enzymes’ substrate/(co)product inhibition as well as unfavorable reaction equilibrium (i.e., for asymmetric amination of ketones) [3,6]. This improvement has been achieved through protein engineering (i.e., structurally-guided or directed evolution) of novel ω-TAs possessing a broader substrate scope [9,10], as well as the development of displacement strategies for the shift of reaction equilibria [11,12,13,14,15]. Therefore, ω-TA-catalyzed reactions have been implemented in industry, and current prominent applications entail (chemo)enzymatic cascades and processes with immobilized biocatalysts in batch or flow reactors [3,6,9,16,17,18,19,20,21,22,23,24].

As mentioned above, recurrent challenges in ωTA-catalyzed reactions using the “classical” alanine (or another α-amino acid) as the sacrificial amine donor in an aqueous environment are unfavorable reaction equilibrium and frequent enzyme inhibition by the amine product and pyruvate co-product [11,25,26,27,28,29,30,31,32]. However, our group and others have demonstrated that these limitations can be addressed (or at least mitigated) by performing the transamination reaction in a neat organic solvent at controlled water activity (a_w_) and using inexpensive 2-propylamine as the amine donor [33,34,35]. Biocatalyst recyclability and long-time stability under process conditions are also critical factors for the successful implementation of enzymes such as ω-TAs in industry, particularly in continuous flow processes [36]. In this context, the development and application of immobilization strategies has enabled enzyme recyclability, thereby enhancing biocatalysts’ performance in both batch and flow reactors [37,38,39,40,41]. A number of these immobilization techniques have been applied on ωTAs, namely entrapment, cross-linking and support binding [17,18,19,20,21,22,23,42,43,44,45,46,47,48,49,50]. Our group recently reported a highly efficient kinetic resolution of a racemic α-chiral amine in both batch and continuous flow using ω-TAs immobilized on polymer-coated controlled porosity glass beads (EziG) [51]. Notably, the same immobilized EziG^3^-ωTA biocatalyst also led to excellent activities for the asymmetric synthesis of α-chiral amines from prochiral ketones in neat organic solvents when the system’s water activity (a_w_) was controlled with hydrate salt pairs [35]. In particular, we obtained conversions of 80% or up to 96% for the asymmetric synthesis of (*R*)-phenoxy-2-propylamine from phenoxyacetone by using only one or two equivalents of 2-propylamine as the amine donor, respectively. Thus, we could increase the substrate concentration from 50 mM to 400 mM without any apparent loss in catalytic performance. The use of the immobilized EziG^3^-ωTA in an organic solvent as a reaction medium did not negatively affect—and possibly even increased—biocatalyst stability, as the reaction could be conducted in a continuous flow reactor for several days.

Nature benefits from a large number of chemically interconnected processes that simultaneously occur in the same living systems. Concurrent artificial catalytic processes inspired by such systems have been developed for the synthesis of enantioenriched compounds [52,53,54,55,56]. Some of these cascades entail strictly parallel biocatalytic reactions; an interesting example is parallel interconnected asymmetric kinetic transformation (PIKAT), in which two redox reactions are generally combined in one-pot in an anti-parallel fashion. Thus, one enzyme—or two enzymes operating in tandem—convert two reagents into two or more useful products via internal cofactor recycling [52,56]. PIKATs were first reported for the asymmetric synthesis of vicinal diols or β-haloalcohols combined with an anti-parallel kinetic resolution of racemic secondary alcohols; the process was catalyzed by an alcohol dehydrogenase (ADH) that required nicotinamide adenine dinucleotide as cofactor (NAD(P)^+^/NAD(P)H) [57,58]. An extension of the approach combined either an asymmetric sulfoxidation or a Baeyer-Villiger oxidation with an anti-parallel kinetic resolution of secondary alcohols [59,60]; in this case, two enzymes were required, namely a Baeyer-Villiger monooxygenase (BVMO) and ADH, which internally recycled the NADP cofactor. One PIKAT example was also reported with an ωTA via internal recycling of the PLP cofactor; in this case, the PIKAT was incorporated into a slightly longer cascade whereby L-threonine was converted to L-homoalanine in an anti-parallel combination with the kinetic resolution of racemic α-chiral primary amines [61].

In this work, we describe a novel PIKAT using an EziG^3^-immobilized and (*R*)-selective ωTA from *Arthrobacter* sp. (EziG^3^-AsR-ωTA) in a neat organic solvent. The immobilization of AsR-ωTA on EziG controlled porosity glass beads was previously investigated and exhibited high retention of enzyme activity and excellent stability in both aqueous and organic media [35,51]. Herein, the same biocatalyst preparation was employed in the organic solvent for the one-pot asymmetric formal reductive amination of a prochiral ketone combined with an anti-parallel kinetic resolution of a racemic α-chiral primary amine (Scheme 1). Due to the more favorable thermodynamic equilibrium for such a PIKAT in the non-aqueous medium, ketone substrates could be combined with racemic amine donors in a 1:2 molar ratio. In a previous study using EziG^3^-AsR-ωTA, toluene was found to be the best solvent when the system’s water activity (a_w_) was fixed and controlled at 0.7 using salt hydrate pairs [35]. Therefore, the same reaction conditions were applied in this study. Over time, the racemic amine (*rac*-**4**–**6b**) undergoes resolution to yield the ketone by-product with a theoretical maximum conversion of 50% and, in principle, leaving the (*S*)-configured amine enantiomer untouched; in the same pot and anti-parallel direction, the prochiral ketone substrate (**1**–**3a**) gets converted to an enantiopure (*R*)-configured amine product.

## 2. Results

### 2.1. Biocatalyst Preparation

EziG^3^ (Fe Amber) beads were used as immobilization carrier material. These beads are made of high porosity glass with pores of well-defined size and geometry (pore diameter: 30 ± 5 nm; pore volume: 1.8 mL g^−1^). The beads’ glass surface (internal and external) is coated with a semi-hydrophobic polymer (total bead’s bulk density: 0.21–0.25 g mL^−1^), which is further derivatized to chelate Fe(III) cations. These Fe(III) centers enable the chemical coordination binding between the enzyme and the carrier [45]. Enzyme loading capacity as high as 30% *w w*^−1^ was reported with this type of carrier material, due to the high surface area of the porous glass per unit of volume [62]. The advantages of metal-ion affinity binding compared with other methods for enzyme immobilization onto a support are: (1) higher enzyme loading capacity; (2) enhanced retention of catalytic activity and stability due to a more oriented binding; (3) possibility to detach the deactivated enzyme, thereby enabling carrier recycling. The other immobilization methods onto a support might have the advantage of requiring less expensive carrier materials (e.g., polymeric materials).

We previously reported the immobilization of an (*R*)-selective ωTA from *Arthrobacter* sp. on EziG^3^ (forming EziG^3^-AsR-ωTA), through the Fe(III)-ion affinity binding between the enzyme and the carrier material. In practice, a genetically fused and N-terminal poly-histidine (His-6) tag of the ωTA interacts with high affinity with the Fe(III) ions onto the polymeric surface of the EziG beads. We previously employed EziG^3^-AsR-ωTA with an enzyme loading of 10% *w w*^−1^ (ratio of enzyme to bead), and quantitative enzyme immobilization was achieved within 2 h incubation time [51]. In this work, we applied the same immobilization conditions, namely AsR-ωTA (2 mg, purified enzyme), EziG^3^ (Fe Amber) beads (20 mg) in KPi buffer (1 mL, 100 mM, pH 8.0) supplemented with PLP (0.1 mM). Under the optimized aqueous reaction conditions (HEPES buffer, pH 7.0, 250 mM; PLP 25 µM), the specific catalytic activity was determined for the transamination of *rac*-methylbenzylamine **6b** (100 mM) with sodium pyruvate (50 mM) at 30 °C [51]. Purified free AsR-ωTA exhibited an activity of approximately 800 U µmol^−1^_enzyme_, whereas immobilized EziG^3^-AsR-ωTA exhibited an apparent activity of approximately 380 U µmol^−1^_enzyme_. The lower apparent activity of the immobilized biocatalyst must be, at least in part, attributed to additional mass transfer limitations in heterogeneous catalysis (i.e., solid supported enzyme in a liquid medium); this is particularly valid for the kinetic resolution of *rac*-methylbenzylamine that has an elevated reaction rate in a homogeneous aqueous system.

### 2.2. PIKAT Substrate Selection

For screening of the PIKAT with the immobilized AsR-ωTA, we selected three prochiral ketones and three racemic α-chiral primary amines comprising aliphatic and aromatic substrates (Scheme 1). Thus, prochiral ketone substrates **1**–**3a** were combined with racemic amine donors **4**–**6b** in a 1:2 molar ratio. Toluene was used as the reaction solvent, and the water activity (a_w_) was controlled using a hydrate salt pair (i.e., Na_2_HPO_3_•5H_2_O/Na_2_HPO_4_•7H_2_O, 1:1 *w w*^−1^), as reported in our previous study [35]. AsR-ωTA—a recombinant (*R*)-selective enzyme originated from *Arthrobacter* sp.—was immobilized on EziG^3^ (Fe Amber) controlled porosity glass beads and prepared for reactions in organic solvent, as previously reported [35,51]. The reactions were performed in dark glass vials (2 mL) with screw caps and shaken in a thermomixer at 25 °C.

Initial experiments indicated the formation of imine side products in the final reaction mixture; these imines arose from the reaction between ketones and amines that were present as substrates and products in the reaction mixture. In fact, the (reversible) imine formation between ketones and amines is more favored in an organic solvent than in an aqueous buffer. Due to the complexity of the obtained reaction mixture, quantification of conversion and enantiomeric excess (%*ee*) of the final products proved difficult. Therefore, we decided to modify the established work-up procedure for enzymatic transamination reactions such that ketones and amines could be separately extracted from the mixture and analyzed in two discrete steps. Although this work-up procedure would require improvements for a preparative or industrial scale application, possibly in a continuous process, separation of the products allowed us to quantify conversions and *ee* by adding an internal standard (IS). Conversions were determined according to the gas chromatography (GC) peak area ratios between the products and IS in the different extraction phases. *ee* values were obtained after derivatization to acetoamido and analyzed using a chiral GC column.

### 2.3. PIKAT Screening of Prochiral Ketones and Racemic Amines

Prochiral ketones **1**–**3a** were successfully employed in the PIKAT using *rac*-**4**–**6b** as amine donors. The reactions were performed at 25 °C in toluene at a_w_ of 0.7 (Scheme 1). The work-up procedure entailed extraction with an HCl aqueous solution followed by treatment with a basic aqueous solution of KOH.

In contrast, for PIKATs with ketone **2a**, the work-up had to be performed using K_2_CO_3_ instead of KOH in the basic extraction step to avoid ester hydrolysis of the final amine product **2b**. The results are shown in Figure 1 (see details in Appendix A, Table A1), wherein each data is the average value obtained from two independent reactions.

The PIKAT of ketone **1a** led to variable conversion levels for the formal reductive amination step (from 48% to 90%) depending on the amine donor (Figure 1a); however, the amine product (*R*)-**1b** was always obtained with >99% *ee*. The conversion of **1a** into (*R*)-**1b** and the outcome of the anti-parallel kinetic resolution of *rac*-**4**–**6b** were obviously mutually influenced; in particular, kinetic resolution theoretically could not exceed the 50% value, as the prochiral ketone and the racemic amine were always applied in a 1:2 molar ratio. In contrast, the kinetic resolution of *rac*-**6b** afforded an apparent conversion of 51 ± 1%, although the ketone **1a** was converted into (*R*)-**1b** with 90 ± 7% conversion. This apparent discrepancy must be attributed to the probably higher volatility of *rac*-**6b**, thereby resulting in an apparent slightly higher formation of acetophenone **6a**. Furthermore, the unreacted amine donor **6b** resulted in 80% *ee* for the (*S*)-configured enantiomer, thus indicating that both enantiomers can be accepted by the AsR-ωTA during the kinetic resolution. The imperfect enantioselectivity of the kinetic resolution step was a general trend, as amines **4b** and **5b** were obtained in average values of 40% and 2% *ee*, respectively.

The PIKAT of ketone **2a** yielded higher conversion for the asymmetric formal reductive amination step such that (*R*)-**2b** was obtained with conversion between 77% and 91% (Figure 1b). Notably, the amine product for the amination of this ketone substrate was also always obtained in >99% *ee*. Analysis of the composition of the reaction mixtures revealed that the kinetic resolutions of the amine donors *rac*-**4**–**6b** additionally yielded enantioenriched mixtures of the (*S*)-configured enantiomers at the end of the reaction (between 64 ± 1% and 80 ± 1% *ee*).

The best performing PIKAT was observed with ketone **3a**; the asymmetric formal reductive amination step afforded quantitative conversion for all of the three investigated cases (Figure 1c). As previously observed for ketones **1a** and **2a**, the amination yielded the amine product (*R*)-**3b** in optically pure form (*ee* >99%). As observed in the previous cases, AsR-ωTA accepted both enantiomers of the racemic amine donors with a preference for the (*R*)-configured ones, thereby leading to enantioenriched mixture of (*S*)-**4**–**6b** at the end of the process (between 52 ± 2% and 59 ± 9% *ee*).

### 2.4. Time Studies

In order to examine the progress of the PIKAT reaction, we performed a time study with ketone **1a** and racemic amine donor **6b**. The progress of the conversion (Figure 2a) and enantiomeric excess (*ee*, Figure 2b) were monitored throughout the course of the reaction (see Appendix A, Table A2 for details). The asymmetric formal reductive amination step of **1a** gradually proceeded until 90 ± 7% conversion; concomitantly, *rac*-**6b** was converted into acetophenone (**6a**) until 50 ± 1% conversion.

Figure 2b shows that the enantioselectivity of the formal reductive amination step remained perfect throughout the course of the reaction (>99% *ee*), whereas the *ee* of the amine donor mixture gradually increased from 0% to ca. 80% (*S*) at the end of the reaction. It is evident that the *ee* of (*S*)-**6b** increased more rapidly at the beginning of the PIKAT, as (*R*)-**6b** is virtually the only accepted enantiomer by the AsR-ωTA at this stage; however, the *ee* curve flattens after 20 h reaction time, thus indicating that a considerable portion (*S*)-**6b** eventually starts to be accepted by AsR-ωTA. This behavior results in the imperfect selectivity of the anti-parallel kinetic resolution step.

### 2.5. Further Studies

Since the PIKAT is based on a reversible amine transfer, we decided to study the influence of the molar ratio between prochiral ketone and a racemic amine donor on the overall thermodynamic equilibrium of the process. Therefore, we decided to increase the prochiral ketone substrate concentration from 50 mM up to 100 mM while keeping the racemic amine concentration constant at 100 mM. Thus, we could check whether an excess of ketone (i.e., from >1 up to 2 eq. in this study) could push the anti-parallel kinetic resolution of the racemic amine to completion at a given ketone substrate concentration. Crucially, high ee values could also be accessible for the kinetic resolution of the racemic amine donor, while the formal reductive amination reaction would always provide >99% *ee* of the desired amine product. Ketone **1a** and amine donor *rac*-**6b** were chosen as model substrates, and the results are reported in Table A3 of Appendix A. These data clearly show that altering the molar ratios between prochiral ketone and racemic amine for the PIKAT did not influence the outcome of the kinetic resolution step, as (*S*)-**6b** was always obtained with an *ee* around 70% when **1a** was added from 50 mM to 80 mM concentration. Moreover, increasing the concentration of **1a** up to 90 mM and 100 mM was detrimental for the selectivity of the kinetic resolution, as the *ee* of (*S*)-**6b** decreased to average values of 48% and 28%, respectively. In contrast, as observed in all this study’s experiments, the asymmetric formal reductive amination step proceeded with excellent enantioselectivity such that (*R*)-**1b** was always obtained in >99% *ee*.

In another set of reactions, we either doubled the enzyme concentration in the reaction or added another portion of fresh enzyme after 24 h reaction time. No improvements were observed (data not shown).

## 3. Discussion

Prochiral ketone substrates **1**–**3a** were successfully employed with racemic amine donors **4**–**6b** in a parallel interconnected kinetic asymmetric transformation (PIKAT). The asymmetric synthesis step yielded (*R*)-configure amines **1**–**3b** with average conversions from 48% to >99% and virtually perfect enantioselectivity (>99% *ee*). Notably, ketone **3a** was quantitatively converted in the three investigated cases and amination of ketone **2a** also led to elevated conversions (77% or higher), whereas ketone **1a** was converted with slightly lower conversions (average values from 48 to 90%) (Figure 1; see Appendix A, Table A1 for details).

Overall, the asymmetric formal reductive amination step appeared to be only minorly affected by the type of racemic amine donor that was applied. The anti-parallel kinetic resolution step showed a consistent trend depending on the applied amine donor; for instance, *rac*-**6b** was always converted to higher conversions compared with *rac*-**4b** and *rac*-**5b** when combined with the same type of ketone in the amination step. In the combination between **1a** and *rac*-**5b**, the kinetic resolution showed a low conversion (19 ± 3%) and nearly racemic amine was obtained (*ee* 2%); accordingly, the concurrent asymmetric formal reductive amination step led to moderate conversion, albeit with an excellent *ee* of >99% (Appendix A, Table A1, entry 2). As the conversion in the kinetic resolution step varied from moderate to high and the enantiomeric excess values were lower than expected, we postulated that the AsR-ωTA is less enantioselective for the kinetic resolution step, and thus, over time, both enantiomers got transformed. The time study of **1a** with *rac*-**6b** (Figure 2a) showed that both asymmetric synthesis and kinetic resolution went towards completion. Compound **1a** was converted to (*R*)-**1b** with 90 ± 7% conversion after 72 h of reaction time, whereas the *rac*-**6b** concentration gradually decreased to reach 50 ± 1% after 72 h. The *ee* percentage values displayed in Figure 2b demonstrate excellent selectivity for the asymmetric formal reductive amination, such that exclusively (*R*)-**1b** was formed. Furthermore, the originally racemic **6b** became an enantioenriched mixture of the (*S*)-enantiomer over time, as mainly the (*R*)-enantiomer was converted to **6a**. After 72 h, the (*S*)-enantiomer comprised approximately 90% of **6b**.

In a subsequent set of experiments, we attempted to increase the conversion and thus the *ee* in the kinetic resolution of *rac*-**6b** by adding more of the prochiral ketone **1a** to the reaction. In this manner, we would sacrifice some conversion in the asymmetric synthesis of (*R*)-**1b** while aiming to gain a higher enantioenrichment of (*S*)-**6b**. Unfortunately, the use of an excess of **1a** (up to equimolar with *rac*-**6b**) did not provide any evident improvement of the reaction outcome (see Appendix A, Table A3 for details). As expected, conversion of **1a** to (*R*)-**1b** was slightly lower than in the previous experiments (Appendix A, Table A1, entry 3) and further decreased while increasing the concentration of **1a**. With the exception of Table A3, entries 5 and 6, all of the reactions exhibited ca. 36-40 mM product formation of (*R*)-**1b** from **1a** (Appendix A, Table A3). Higher concentration of **1a** (90 and 100 mM; Appendix A, Table A3, entries 5 and 6) hampered the PIKAT such that the average productivity of (*R*)-**1b** was 31 mM and 20 mM, respectively. We attribute this effect to the known ωTA enzyme’s inhibition at higher concentrations of some ketone substrates [11,26,28,29,30,31,32]. On the other hand, kinetic resolution of *rac*-**6b** was not improved by increasing the concentration of **1a**, as the *ee* values never exceeded 78 ± 2% (Appendix A, Table A3) and never improved compared with the value of 80 ± 0% reported in Appendix A, Table A1 entry 3. Finally, doubling the amount of immobilized enzyme or adding fresh enzyme after 24 h of reaction time did not benefit the reaction outcome.

## 4. Materials and Methods

### 4.1. Chemicals and Carrier Materials

Ketones **1**–**3a**, racemic amines **4**–**6b** and pyridoxal-5′-phosphate (PLP) were purchased from Sigma-Aldrich (Steinheim, Germany). EziG^3^ (Fe Amber) enzyme carrier material was provided by EnginZyme AB (Solna, Sweden). Further details for equipment and analytical determination are presented in Section 4.6. All reaction solvents were degassed before use. All of the water-equilibrated solvents were prepared by shaking hydrate salt pairs in the organic solvent for 1 h at RT.

### 4.2. Expression and Purification of ωTAs

C-terminal His-tagged (*R*)-selective ωTA from *Arthrobacter* sp. (AsR-ωTA) [63,64] was expressed and purified as previously reported [51].

### 4.3. Bradford Assay

Protein assay dye reagent concentrate (BioRad Laboratories, Veenendaal, The Netherlands) was diluted 5 times with MilliQ water and filtered through a paper filter. This diluted stock solution was freshly prepared before use and kept in the dark at 4 °C. The albumin calibration line was determined in the standard range of 200–1000 µg mL^−1^ protein concentration according to the supplier-provided protocol. A low-concentration assay was used in cases of lower protein concentrations (<25 µg mL^−1^). Samples were prepared by mixing 980 µL diluted stock solution and 20 µL protein sample (for low-concentration assay: 800 µL diluted stock solution and 200 µL protein sample) followed by incubation for 5–10 min at room temperature. Absorption at 595 nm was measured and plotted against the protein concentration. Diluted ωTA enzyme samples were then prepared and measured in the same fashion in order to determine their concentration using the albumin calibration line.

### 4.4. Immobilization of AsR-ωTAs on EziG^3^ Carrier Materials

EziG^3^ carrier material (20 ± 0.2 mg) was cooled down in an ice bath and suspended in an immobilization buffer (KPi, 1 mL, 100 mM, pH 8.0) supplemented with PLP (0.1 mM). Purified ω-TA (2 mg, equal to 10% *w w*^−1^; enzyme loading to support material) was added to the suspension. The mixture was shaken on an orbital shaker (120 rpm) for 3 h at 4 °C. Small aliquots from the aqueous phase (20 μL) were taken before and after the immobilization procedure, and their concentrations were determined using the Bradford assay (see Section 4.3 for details). Once full immobilization was obtained, the immobilized enzyme was let to sediment and the buffer was removed by pipetting. We recovered the immobilized enzyme by sedimentation (1 min), since the use of other separation techniques such as centrifugation or vacuum filtration can lead to partial enzyme deactivation.

### 4.5. PIKAT in Organic Solvents with Immobilized ωTAs

In a dark glass vial (2 mL), immobilized EziG^3^-AsR-ωTA (total mass 22 mg, 10% *w w*^−1^ enzyme loading to support material) and hydrate salt pair (Na_2_HPO_3_•5H_2_O/Na_2_HPO_4_•7H_2_O, total mass 40 mg, 1:1, *w w*^−1^) were suspended in EtOAc (1 mL, at fixed a_w_ of 0.7) and shaken for 15 min (900 rpm, thermomixer). The EziG^3^-AsR-ωTA was allowed to sediment and the solvent was removed by pipetting. The EziG^3^-AsR-ωTA with hydrate salt pair was resuspended in another aliquot of EtOAc, and the above-described process was repeated twice. Next, the EziG^3^-AsR-ωTA with hydrate salt pair was washed with toluene (1 mL, at fixed a_w_ of 0.7) and then allowed to sediment, after which the solvent was removed by pipetting. Finally, toluene (1 mL total reaction volume, at fixed a_w_ of 0.7) was added as a reaction solvent. The ketone substrate **1**–**3a** (final concentration: 50 mM) and racemic amine donor **4**–**6b** (final concentration: 100 mM) were added, and the reaction vials were shaken in an upright position (900 rpm, thermomixer) for 72 h at 25 °C. Work-up was performed by separating ketone and amine compounds from each other by extraction. First, the EziG^3^-AsR-ωTA was left to sediment and the organic reaction mixture was separated from the biocatalyst by pipetting. Then, naphthalene was added as an internal standard to the organic reaction mixture from a concentrated stock solution (50 µL from 1 M stock in toluene, final concentration 50 mM). The amine compounds were selectively extracted from the toluene reaction phase using an aqueous 2 M HCl solution (400 µL), thereby minimizing the possible extraction of ketones. Following mixing and centrifugation (14 krpm, 5 min, 4 °C), the two layers were separated and the organic layer was kept separate. The acidic aqueous layer was re-extracted with toluene (2 × 500 µL) to selectively remove potential traces of ketone. The organic layers were combined and dried over MgSO_4_ prior to analysis. Then, the acidic aqueous layer—containing the protonated amine products—was basified using an aqueous solution of either KOH (200 µL of a 10 M stock solution) or K_2_CO_3_ (for reactions with substrate **2a**), and extracted with EtOAc containing 50 mM naphthalene as an internal standard (2 × 500 µL). Drying over MgSO_4_ was performed prior to analysis. Analysis was performed by separately injecting ketone- and amine-containing solutions on GC equipped with an achiral column (see Section 4.6 for details on analytical equipment and determination). For the determination of enantiomeric excess, the samples containing amines were derivatized by incubation with 4-dimethylaminopyridine in acetic anhydride (final concentration: 5 mg mL^-1^) for 30 min (170 rpm, RT). Samples were quenched by the addition of water (500 μL) and shaken for 30 min (170 rpm, RT). Following centrifugation, the organic layer was dried over MgSO_4_ and analyzed by GC equipped with a chiral column (see Section 4.6 for details on analytical equipment and determination).

### 4.6. Analytical Equipment and Determination

The biotransformation conversions were determined using a 7890A GC system (Agilent Technologies, Santa Clara, CA, USA) equipped with a FID detector with H_2_ as carrier gas and a DB-1701 column from Agilent (30 m, 250 μm, 0.25 μm), or a HP-5 column from Agilent (30 m, 320 μm, 0.25 μm). Chiral samples were measured using a CP-DEX column from Agilent (25 m, 320 μm, 0.25 μm) or a HydroDex-β-TBDAc column (50 m, 250 μm) from Macherey-Nagel (Dueren, Germany):DB1701-30 m-method-A: constant pressure 6.9 psi, T injector 250 °C, split ratio 50:1, T initial 80 °C, hold 6.5 min; gradient 10 °C/min up to 160 °C, hold 5 min; gradient 20 °C/min up to 200 °C, hold 2 min; gradient 20 °C/min up to 280 °C, hold 1 min.DB1701-30 m-method-B: constant pressure 6.9 psi, T injector 250 °C, split ratio 50:1, T initial 60 °C, hold 6.5 min; gradient 20 °C/min up to 100 °C, hold 1 min; gradient 20 °C/min up to 280 °C, hold 1 min.HP-5-method-A: constant pressure 4 psi, T injector 250 °C, split ratio 30:1, T initial 60 °C; gradient 5 °C/min up to 150 °C, hold 1 min; gradient 10 °C/min up to 250 °C, hold 1 min.HP-5-method-B: constant pressure 4 psi, T injector 250 °C, split ratio 30:1, T initial 40 °C, hold 2 min; gradient 5 °C/min up to 80 °C, hold 2 min; gradient 20 °C/min up to 250 °C.CP-DEX-method-A: constant flow 1.4 mL/min, T injector 200 °C, split ratio 40:1, T initial 100 °C, hold 2 min; gradient 1 °C/min up to 130 °C, hold 5 min; gradient 10 °C/min up to 170 °C, hold 10 min.; gradient 10 °C/min up to 180 °C, hold 1 min.CP-DEX-method-B: constant flow 1.4 mL/min, T injector 200 °C, split ratio 40:1, T initial 100 °C, hold 2 min; gradient 1 °C/min up to 118 °C, hold 5 min; gradient 10 °C/min up to 170 °C, hold 10 min.; gradient 10 °C/min up to 180 °C, hold 1 min.CP-DEX-method-C: constant flow 1.5 mL/min, T injector 200 °C, split ratio 20:1, T initial 60 °C, hold 2 min; gradient 5 °C/min up to 100 °C, hold 2 min; gradient 10 °C/min up to 180 °C, hold 1 min.HydroDex-β-TBDAc-method: constant flow 1 mL/min, T injector 220 °C, split ratio 20:1, T initial 100 °C, hold 2 min; gradient 1 °C/min up to 150 °C, hold 8 min; gradient 10 °C/min up to 170 °C, hold 1 min.

## 5. Conclusions

In this work, we report the successful development of the first example of parallel interconnected kinetic asymmetric transformation (PIKAT) using an immobilized ω-transaminase in a neat organic solvent. PIKATs have been achieved almost exclusively using oxidoreductase enzymes, and other enzyme classes such as transferases have rarely been considered. The PIKAT with ωTA entails an asymmetric formal reductive amination of a prochiral ketone combined with a concomitant and anti-parallel kinetic resolution of a racemic amine. The use of a neat organic solvent is critical for the successful application of the PIKAT concept with ωTA, as it enables the appropriate tuning of the thermodynamic equilibrium between two anti-parallel reactions, also using structurally diverse substrates. The formal reductive amination step always proceeded with excellent selectivity, thus yielding the (*R*)-configured amine product in >99% *ee*. Conversely, the immobilized ωTA from *Arthrobacter* sp. displayed only moderate selectivity in the kinetic resolution step, thereby resulting in a maximum of 80 ± 1% *ee* value for the less converted (*S*)-configured amine donor. Time studies showed that the kinetic resolution step proceeded with main (or even sole) consumption of the (*R*)-configured amine until a conversion of ca. 35% was obtained, at which stage it appeared that the other enantiomer started to be accepted as well, ultimately leading to a 51 ± 1% conversion and 80% *ee*. In contrast, the anti-parallel formal reductive amination proceeded smoothly until 90 ± 7% conversion, and >99% *ee* of the final amine product was obtained. The enantioselectivity of the kinetic resolution step can potentially be enhanced by testing other available ωTAs and/or engineering new and more selective variants that are customized for this process. Admittedly, one limitation of the current process is the elaborated work-up procedure entailed in the selective extraction of the amine products (and ketone by-product) from the reaction mixture. Therefore, further work is required to develop more efficient procedures for the isolation of pure products, such as using fractional distillation and/or selective ion-exchange or hydrophobic resins [27]. In this context, strategies for selective product removal could be implemented along with PIKAT running in continuous flow reactors. In summary, this work provides a proof-of-concept for such a PIKAT process, which was notably applied with a selected range of structurally diverse ketones and racemic amines that comprise aliphatic, aromatic and arylaliphatic substrates.

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
