# Peer review of "Parallel Interconnected Kinetic Asymmetric Transformation (PIKAT) with an Immobilized ω-Transaminase in Neat Organic Solvent"

_molecules, 2020, doi:10.3390/molecules25092140_

Round 1

Reviewer 1 Report

The work deals with the use of an immobilized w-transaminase aimed to the preparation of an optically pure amine starting from the corresponding ketone, in an organic solvent.

In particular, the process is a parallel interconnected kinetic asymmetric transformation (PIKAT). The authors recently published an article about the same topic (Advanced Synthesis and Catalysis, 2020, 362, 1-11) studying the PIKAT between only one ketone and only one amine. In this article they considered three different types of carrier material, focusing their attention on the choice of the solvent, on the pH effect and on the water activity.

In the present manuscript they describe the same PIKAT applied to different ketones (two new examples) and amines (three new examples) together with a time study (partially reported also in the previous article) and an investigation about the effects of different molar ratios.

For these reasons the manuscript is acceptable for publication on ‘’Molecules” but the authors must clearly underline the novelty of the present work. In the Abstract (line 22-23) they cannot define the PIKAT as ‘’the first reported’’ one.

Minor Comments: A Scheme describing the literature examples (lines 73-90) of PIKAT should be added.

(S)-configured amine should be changed into (S)-amine in the entire text.

Why the Tables are added as an Appendix? Only Table1 is named Tables 1A

Author Response

Please see the attachment, for scientific reply.

Regarding English language and style, I want to point out that the previous version of the manuscript already underwent professional proof-reading by native English speakers.  Therefore, the comment from Reviewer 1 on this regard is unreliable in our opinion. We also assume that Reviewer 1 is not a native English speaker from the language and style used in his report.

Reviewer 2 Report

The experimental work appears to have been carried out well. However, some points deserve attention for further publication in Molecules journal. I suggest that it is accepted for publication after the following revisions:

What is the nature of the interaction between the enzyme and the carrier? The authors should clarify this point in the manuscript.

Line 119. Essential data for an enzyme immobilization paper is missing.  What is the immobilization yield? What is the expressed activity?

What is the loading capacity of the support?

Line 284. How was the enzymatic activity of the heterogeneous biocatalyst determined? This information is essential to determine if the immobilization affects the enzyme activity.

Line 290-291. The heterogeneous biocatalyst was recovered by sedimentation. A question arises in this regard, why was the biocatalyst not centrifuged or vacuum filtered for recovery?

Author Response

Please see the attachment, for scientific reply.

Regarding English language and style, we corrected a couple of typos and adjusted one sentence in the conclusion.

Reviewer 3 Report

In this interesting study, a novel Parallel Interconnected Kinetic Asymmetric Transformation (PIKAT) by using a microbial ω-Transaminase immobilized on an inorganic support has been reported. This biocatayst has been used in the asymmetric formal reductive amination of a prochiral ketone combined with a concomitant and anti-parallel kinetic resolution of a racemic amine in a solvent system. The paper is relevant as content, with high degree of novelty and logically built.

a) A large variety of organic and inorganic supports has been succesfully applied in the preparation of active and stable biocatalysts. In this study, controlled porosity glass beads was used. The authors should briefly describe the motivation for preparing heterogeneous biocatalysts using inorganic supports. Moreover, better explain the protocol used to prepare this biocatalyst (covalente attachment, adsorption, or entrapment). Some features of this support are required (particle size, surface area, pore size, etc...)

b) The authors should perform assays using the free enzyme to show the relevant effect of the immobilization step on the activity and selectivity of the reaction.

c) Reusability tests should be performed after successive cycles of reaction.

Author Response

Please see the attachment, for scientific reply.

Regarding English language and style, I want to point out that the previous version of the manuscript already underwent  to professional proof-reading by native English speakers.  A couple of typos and one sentence was modified in the conclusion.

Round 2

Reviewer 3 Report

The authors corrected the manuscript,as suggested. I recommend acceptance for publication in this journal.